# RecSpy: Cognition-Driven PIN Inference on Randomized Soft Keyboards

## Abstract

As mobile devices have become deeply integrated into daily life, users often input sensitive data (i.e., PINs) to unlock services or authorize payments, which introduces high risks of side-channel attacks. To defend against potential attacks, in practice, soft keyboards for PIN entry are randomized in layout to mitigate such threats. In this paper, we present *RecSpy*, a novel cognition-driven acoustic side-channel attack that infers PINs on randomized soft keyboards. Unlike prior work that relies on video, power, or electromagnetic emanations, RecSpy exploits a previously **unexplored vulnerability:** distinct human recognition latencies for symbolic numbers. By modeling cognitive latency patterns and leveraging acoustic keystroke signatures, RecSpy learns individual and digit-level recognition features through contrastive and self-supervised learning. Furthermore, we also introduce a novel **Logic-Guided Inference Network** that integrates recognition patterns with the reasoning capabilities of a large language model (LLM) to prune the hypothesis space and infer complete PIN sequences. We extensively evaluate RecSpy on both Android and iOS devices, and results show that it improves the probability of successful inference by up to 4000×, which demonstrates a practical threat to current mobile authentication systems and shows that representation learning and LLMs can enable new side-channel attacks.

## 1 Introduction

As mobile devices have become deeply integrated into daily life, users frequently enter highly sensitive data, such as card security codes or personal identification numbers (PINs). For instance, a smartphone user may unlock a banking app with a six-digit PIN before authorizing a funds transfer. While this common use case improves user convenience, it simultaneously creates an opportunity for attackers to capture sensitive information. In other words, adversaries can exploit through side-channel attacks to steal sensitive information for legitimate users Wang et al. (2019); Cronin et al. (2021); Jin et al. (2021).

To launch these attacks, researchers have introduced several interesting approaches. Specifically, shoulder-surfing-resistant blurred the app UI to bystanders against eavesdropping from side-view or video recording Tang & Shin (2023). Charger-surfing used additional hardware to eavesdrop sensitive data generated by legitimate users Cronin et al. (2021). Periscope leveraged customized antenna and control board to listen EM emissions from smart devices Jin et al. (2021). However, these approaches either assume explicit access to legitimate users' hardware or require additional devices to launch the attack, which makes existing approaches impractical and unstealthy. For example, current smart devices normally have OS-level indicators (e.g., LED lights) that illuminate when the camera is active, which makes camera-based attacks detectable. In addition, recent research has proven that even small physical-layer interference can disrupt signals leaked from smart devices Tang & Shin (2023), which reduces attack success rates. Crucially, *existing attacks target conventional keypads with **fixed layouts***, while modern mobile applications (Apps) have employed randomized layouts Kirkwood et al. (2022), which further reduces their practical feasibility.

Different from existing approaches, we introduce *RecSpy*, a novel side-channel attack that leverages human cognitive recognition patterns for symbolic numbers to infer PINs entered on randomized soft keyboards. Our threat model considers a practical setting where the victim's device hosts a malicious or seemingly benign app embedding third-party libraries with only minimal permissions (e.g., microphone). Such microphone access is extremely common across mobile apps Kaspersky (2025), which makes our attack both realistic and stealthy. In addition, since *RecSpy* only uses cognitive-level differences in numeral recognition, it can be directly applied to various smart devices.

However, to launch the attack, we need to overcome three key challenges: **(i) How to stealthily eavesdrop the keystroke signatures of user inputs?** As camera permission is highly restrictive and any explicit use of hardware is readily noticeable, it is particularly challenging to practically deploy such attacks. To address this, we leverage built-in microphones to eavesdrop the keystroke acoustics without requiring any additional hardware. **(ii) How to obtain an individual's detailed recognition latency for each digit?** Although keystroke signatures can be captured from acoustic recordings, randomized soft keyboard layouts break the direct mapping between a keystroke event and the corresponding digit. This misalignment makes it infeasible to label per-digit latencies directly from raw keystroke traces. To overcome this, motivated by prior findings that individuals exhibit distinct recognition latencies to digits and this characteristic varies with age Nara et al. (2023); Schneider et al. (2017), we design a novel **Individual-level Recognition Mapper (IRM)** and a **Digit-level Discrimination Network (DDN)** to contrastively learn individual-specific patterns and refine them into digit-level latencies. **(iii) How to effectively narrow down the hypothesis space for inference?** Even with a fully learned digit recognition pattern for each individual, PIN inference still remains challenging. This is because subtle recognition differences cannot be effectively captured using traditional classification models Sun et al. (2020). To overcome this challenge, we design an innovative **two-stage reasoning module** that leverages the logical reasoning capability of a large language model (LLM) to utilize extracted patterns and hidden cues for inference enhancement.

The contributions of this paper can be summarized as follows:

• To our knowledge, *RecSpy* is the first side-channel attack that leverages previously-unexplored human cognition patterns for PINs inference on randomized soft keyboards.

• We introduce a novel Individual-level Recognition Mapper (IRM) and a Digit-level Discrimination Network (DDN) to extract individuals' recognition patterns. Additionally, we creatively leverage the reasoning ability of an LLM to automatically extract the hidden semantic information of the keystroke signatures to narrow the hypothesis space for PIN inference.

• We extensively evaluate *RecSpy* in real-world scenarios on both Android and iOS platforms. The experiment results show that *RecSpy* effectively improves the probability of inference by up to 4000 times against the random guessing baseline of $10^{-6}$ for 6-digit PINs.

## 2 RELATED WORK

Researchers have demonstrated keystroke recovery using various external hardware. For example, SpiderMon recovers PINs on physical keypads by analyzing reflected cellular signals Ling et al. (2020); GazeRevealer infers PINs from eye movement reflections Wang et al. (2019). Screen-gleaning reconstructs on-screen content using an antenna and SDR to recover sensitive information Liu et al. (2020). Periscope places an antenna beneath a desk to eavesdrop keystrokes from EM radiation changes Jin et al. (2021). On the other hand, acoustic-based PIN inference has been investigated by researchers. KeyListener uses both speaker and built-in microphone to localize finger taps on a soft keyboard in real, noisy environments Lu et al. (2019). Shumailov solely leverages built-in microphones to capture the propagation characteristics of touch-induced sounds on soft keyboards to infer PIN digits Shumailov et al. (2019).

Although these attacks demonstrate feasibility in some scenarios, they either require external hardware or assume that the attacker can pre-instrument the environment, which is impractical for real-world deployments. More importantly, their attacks are typically demonstrated only on conventional keyboard layouts. **In contrast,** our attack exploits human cognitive recognition patterns and only requires the access to built-in microphones, which is more practical and can be directly applied to randomized-layout soft keyboards.

## 3 BACKGROUND AND OBSERVATION

### 3.1 BACKGROUND: LAYOUT OF BUILT-IN MICROPHONES

In *RecSpy*, we mainly use smartphones to capture keystroke acoustics. Specifically, modern smartphones are equipped with multiple built-in microphones that support ambient noise suppression, echo cancellation, and stereo recording Apple (2025); Developers (2023). As shown in Figure 1, the primary microphone is typically located near the charging port. The front microphone is near the front camera, and certain high-end devices also have rear microphones by the back camera. During stereo recording, these microphones capture acoustic events and generate two-channel signals based

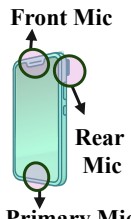

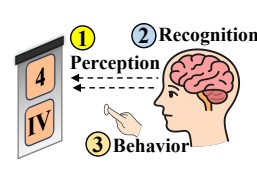

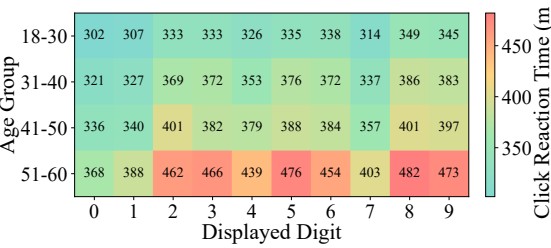

**Figure 1:** The Mic Layout.

**Figure 2:** The Process of Recognition.

**Figure 3:** Digit Recognition Latency (Perception to Behavior) for Individuals Aged 51 to 60.

on inter-channel time and intensity differences for spatial sound source tracking. In this work, we also exploit this stereo recording capability to eavesdrop keystroke sounds.

### 3.2 BACKGROUND: PRINCIPLE OF NUMBER RECOGNITION

As shown in Figure 2, the general process of human number recognition follows a well-established cognitive pathway Dehaene et al. (1993); Zorzi et al. (2011); Pollack & Price (2019). **First,** visual information of symbolic numbers (e.g., Arabic numerals or number words) and non-symbolic numbers (e.g., dot patterns) are processed by the early visual system. **Second,** this information stimulates the occipital visual cortex of the brain Goebel (2012) and is conveyed to the visual word form area (VWFA), which identifies symbolic numbers. **Third,** the recognized digit is processed in the intraparietal sulcus region (IPS), which is associated with the semantic processing of numerical magnitude. **Finally,** the process leads to an observable behavior, such as button pressing. Building on this pathway, researchers have demonstrated that the process of symbolic number recognition is linked to non-symbolic quantity representations, and symbol shape complexity affects recognition latency Piazza et al. (2007); Nieder (2016).

These studies have established two important conclusions: **(i)** different digits require different processing times Nara et al. (2023), and **(ii)** this recognition ability declines with age Schneider et al. (2017). Guided by these findings, we hypothesize that: *recognition latency varies across digits and individuals, and these cognitive differences leave measurable traces in everyday interactions, such as PIN entry.*

### 3.3 OBSERVATION: DIGIT RECOGNITION TIME

To study symbolic number recognition latency, we recruited 143 volunteers evenly distributed across four age groups. Each participant tapped a continuously refreshed randomized keyboard for 50 trials per digit, and the recognition duration was recorded by the smartphone's internal clock, which we denote as **digit recognition latency**. To support later model training, participants also completed full PIN inputs, yielding 4,735 stereo keystroke instances. We preprocessed the 4,735 PIN-entry samples by removing silence and aligning them to 4.41 seconds with zero-padding. Since the resulting per-digit latencies closely matched those obtained from isolated single-digit trials, we show the average digit recognition latencies across age groups in Figure 3.

Overall, younger participants (18 - 30) had the shortest average recognition latency (approximately $328ms$). In contrast, the average recognition latency of older participants is around $440ms$. This difference reflects a clear age-related decline in recognition speed. Across all age groups, digits with simple or distinctive shapes, such as 0, 1, and 7, showed shorter latencies. Meanwhile, visually complex digits, including 8 and 9, had the highest latencies. The corresponding recognition latency can be as high as $482ms$. For participants aged 31 - 40 and 41 - 50, recognition latencies increased gradually, but the relative differences among digits remained unchanged. **In summary,** digit recognition latency increases with age and also depends on digit shape. Across all age groups, digits 0, 1, and 7 are recognized fastest, whereas 8 and 9 are consistently the slowest. These stable latency differences indicate that digits can be reliably distinguished.

## 4 THREAT MODEL

• **System Model.** We consider a smartphone with a capacitive touchscreen and multiple built-in microphones. The device runs common applications that require PIN-based authentication, such as screen unlocking, mobile payments, or account logins. Users enter 6-digit PINs through the system-

provided randomized soft keyboard. Each keystroke produces both visual feedback and an acoustic event captured by the microphones at standard sampling rates (44.1–48 kHz).

● **Threat Model.** Following prior reports that 10%–24% of smartphones have hosted malicious apps at least once Kotzias et al. (2021), we assume the adversary controls a stealthy app installed on the victim's device. This is a reasonable assumption, since users may unknowingly install such software via third-party app stores, phishing links, IM app sharing, or through bundleware from seemingly harmless apps. Once granted microphone permission, it records acoustic signals during PIN entry and uploads them for offline analysis. The adversary's goal is to reconstruct the victim's full 6-digit PIN.

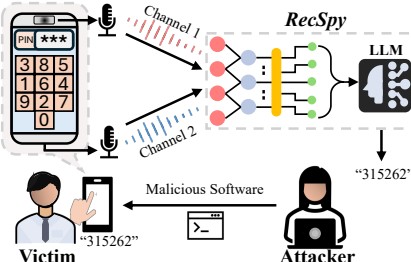

**Figure 4:** Attack Scenarios.

As illustrated in Figure 4, the victim enters a PIN on the randomized soft keyboard, and the built-in microphones capture the resulting keystroke sounds as stereo signals. These signals are intercepted by the malicious app and transmitted to the adversary. *RecSpy* then processes the two-channel acoustic data and leverages an LLM-based reasoning module to infer the most likely PIN sequence.

● **Assumptions.** To ensure the practicality, we assume: **(i)** the victim's device runs an unmodified Android or iOS system; **(ii)** the victim enters PINs as in normal daily use; **(iii)** the adversary has sufficient resources for machine learning–based inference. **Importantly,** we do not assume special privileges such as modifying the keyboard layout or accessing screen buffers, and the attack is not limited to a specific smartphone model.

## 5 DESIGN OVERVIEW

The **design goal** of *RecSpy* is to enable effective PIN inference on randomized soft keyboards. As discussed in Section 1, this requires addressing three challenges: **(i)** capturing keystroke signatures in a stealthy and practical manner, **(ii)** modeling an individual's digit-specific recognition latency, and **(iii)** narrowing the hypothesis space for reliable PIN inference. To overcome these challenges, *RecSpy* consists of three novel modules: **(i)** the Individual-level Recognition Mapper, **(ii)** the Digit-level Discrimination Network, and **(iii)** the Logic-Guided Inference Network.

● **Individual-level Recognition Mapper (IRM).** The IRM module aims to extract an individual's unique recognition pattern from the keystroke signatures. Unlike prior approaches that rely on additional hardware (e.g., cameras, antennas) or privileged OS access, *RecSpy* exploits acoustic signals recorded by built-in microphones, which provides a stealthy and realistic attack vector. However, extracting recognition patterns is challenging: keystroke sounds are weak, easily corrupted by noise, and the differences across digits or users are subtle. To address this, IRM applies spectrogram enhancement and introduces a novel **frequency-dependent weighting component** that focuses on the most informative spectral regions. Moreover, to further magnify the pattern differences across individuals, IRM module is optimized using a contrastive learning strategy. Finally, IRM module generates the individual's recognition pattern for fine-grained digit-level pattern alignment.

● **Digit-level Discrimination Network (DDN).** The DDN module is used to capture digit-level recognition latency patterns for each individual. However, since the recognition latencies are distinct to each digit, it is difficult to make digit-specific modeling for each individual. To overcome this challenge, we introduce an innovative **Attention-based Conditioning Block (ACB)** that aligns an individual's recognition pattern with specific digits. The ACB combines keystroke acoustics with the recognition pattern through **feature-wise linear modulation**. To capture multi-scale digit-level patterns, the DDN further stacks multiple ACBs in sequence. Since detailed latency annotations for individual keystrokes are impractical to obtain, it is hard to train the DDN module. To address this issue, we apply a self-supervised learning strategy to learn effective representations without explicit labels. The resulting DDN produces fine-grained digit-level recognition latency patterns that support accurate PIN inference.

● **Logic-Guided Inference Network (LIN).** The goal of LIN is to narrow the hypothesis space for effective PIN inference. The **core idea** is to leverage the reasoning capability of large language models (LLMs) to incorporate logical cues extracted from keystroke acoustics. Specifically, LIN operates in two stages: **Logic Extractor** and **Pruning Reasoner**. In the Logic Extractor stage,

LIN extracts hidden cues from the keystroke acoustics. To do this, the acoustic spectrogram is first encoded into a sequence of tokens. These tokens are then paired with an instruction template and fed into the LLM. Based on this input, the LLM produces position-wise priors that capture latent recognition patterns. In the Pruning Reasoner stage, these priors are combined with recognition patterns from earlier modules. The combined representation is then used to construct a new instruction template. This template will guide the LLM to prune the hypothesis space and retain only the most likely PIN candidates.

# 6 DETAILED DESIGN

## 6.1 INDIVIDUAL-LEVEL RECOGNITION MAPPER

The **goal** of the **Individual-level Recognition Mapper (IRM)** module is to obtain an individual's recognition pattern from the keystroke acoustics. To achieve this goal, as shown in Figure 5, we design a recognition encoder that can differentiate the individual-specific recognition latencies even from weak and noisy keystroke signals.

First, to mitigate spectral leakage and noise interference in the keystroke acoustics, we introduce a two-layer Spectrogram Enhancement Network (SEN) Yang et al. (2023). The SEN removes leakage artifacts and sharpens frequency components, particularly in the mid-to-high frequency range (e.g., 4–10 kHz).

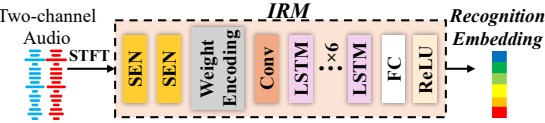

**Figure 5:** The architecture of IRM.

We transformed keystroke acoustics into the two-channel audio spectrogram using Short-Time-Fourier-Transform (STFT) and input the spectrogram into the SENs. By doing this, the SENs restore the corrupted spectral details. These enhanced inputs can help the subsequent layers to discriminate recognition latencies. However, since traditional convolutional neural networks (CNNs) are shift-invariant, spectral components at distinct frequencies are treated as equivalent patterns, which makes frequency-specific details blurred. To address this limitation, we introduce a novel **frequency-dependent weighting component** that assigns each frequency bin with a unique frequency-related weight value. Specifically, the weight of the $i$-th frequency bin can be defined as:

$$\omega_i = \alpha \cdot \frac{\ln\left(1 + \beta \cdot \frac{i}{N-1}\right)}{\ln(1 + \beta)}, \quad i = 0, \ldots, N-1 \tag{1}$$

where $N$ is the number of frequency bins, and $\alpha$ and $\beta$ are tunable parameters ($\alpha = 0.1$ and $\beta = 8$ in our design). This weighting scheme assigns larger weights to mid- and high-frequency bins, where keystroke cues are most prominent. Meanwhile, the other bins receive relatively uniform weights. As a result, the IRM module highlights the spectral regions of interest (ROIs) that carry the most discriminative information. To extract recognition patterns from these ROIs, we first apply a CNN layer to capture local spectral features. The outputs are then passed through a stack of six Long Short-Term Memory (LSTM) layers. Lower layers focus on detecting event boundaries and their associated latencies, while higher layers aggregate these signals into a coherent temporal representation. Finally, the aggregated features are projected into an individual's **recognition embedding**, which serves as the input for digit-level pattern alignment.

**Training Scheme.** To enhance inter-individual separation, we train the IRM module with a supervised contrastive learning strategy. This approach clusters embeddings with similar latency patterns and separates those with different patterns. We use the dataset in Section 3, labeling each instance by age group. To ensure balanced digit distribution, we downsample overrepresented digits, yielding 3,568 instances. In each training step, we sample 8 instances from each of the four age groups to form a batch of 32. Each instance is treated as an anchor: samples from the same age group serve as positives, while those from other groups serve as negatives. All instances are passed through the IRM to obtain recognition patterns, which are projected by a two-layer MLP with ReLU activation. Pattern similarities are optimized with the *SupCon* loss Khosla et al. (2020), using a temperature parameter $\tau = 0.07$ to sharpen the separation between positives and negatives. After training, the IRM generates recognition patterns that are used for fine-grained digit-level pattern alignment.

## 6.2 DIGIT-LEVEL DISCRIMINATION NETWORK

In this section, we design a **Digit-level Discrimination Network (DDN)** module to learn an individual's digit-level recognition latency pattern under self-supervision. Specifically, since the recog-

nition latency is distinct to each digit, it remains challenging to model the digit-level recognition pattern solely from the recognition embedding. In addition, it is impractical to label detailed digit-specific latencies. To overcome these challenges, our DDN module uses a novel **Attention-based Conditioning Block (ACB)** to align the digit-level recognition pattern with the recognition embedding from previous module.

As shown in Figure 6, the ACB is composed of two branches. On the one branch, we adopt a feature extractor that aims to extract digit-level latency features. This feature extractor consists of a frequency transformation block (FTB) followed by max-pooling and a convolution layer. The **key idea** is that the FTB can generate a global time-frequency attention map to have the full-frequency receptive field. Therefore, we use the FTB to capture the global correlations among the time and frequency dimensions. Then, we apply max-pooling to the FTB output to amplify the attention map in the regions corresponding to keystroke events, since these

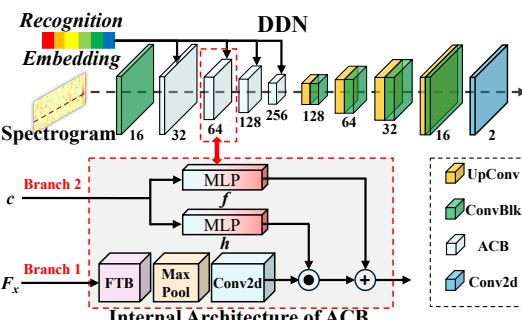

**Figure 6:** The architecture of DDN.

events manifest as spectral-energy peaks. Through this attention map, we can guide the CNN layer to extract the digit-level latencies between these pronounced keystroke events. On the other branch, we aim to personalize the feature extractor to specific individual's recognition pattern. To do this, we use the recognition pattern as a conditional vector to generate feature-scaling and feature-shifting parameters through multilayer perceptrons (MLPs). Then, we apply **feature-wise linear modulation** to the extracted digit-level latency pattern:

$$F_c = f(c) \odot norm(F_x) + h(c) \tag{2}$$

where $F_x$ is the output of the feature extraction, $c$ is the recognition embeddings as the condition features, and $f(c)$ and $h(c)$ are the scaling and shifting parameters. This modulation can amplify discriminative components and suppress noise, which aligns the digit-level latency pattern with the individual's recognition pattern.

To further obtain the fine-grained digit-level pattern, we stack a CNN block and four ACBs with output channels of 16, 32, 64, 128, and 256, respectively. First, the CNN block extracts local spectral–temporal features from the keystroke spectrogram and projects them into an intermediate representation space for subsequent alignment. Then, the stacked ACBs provide multi-scale fusion to enlarge the effective receptive field for better pattern alignment, while simultaneously ensuring sufficient capacity for representation learning without introducing unnecessary complexity.

**Training Scheme.** Since the detailed annotation for digit latencies is unrealistic, to effectively optimize the DDN module for pattern alignment, we employ a decoder to apply self-supervision. The ACB-stacked encoder extracts compact latent representations of digit-level recognition latency patterns, while the decoder reconstructs the original spectrogram from these latent features, thereby enforcing the preservation of fine-grained patterns. We train DDN with an L1 reconstruction loss $\mathcal{L}_{L1}$ on the dataset from Section 3.3 for 80 epochs. After training, DDN produces individualized digit-level latency patterns for each user.

### 6.3 Logic-Guided Inference Network

The **goal** of the **Logic-Guided Inference Network (LIN)** is to narrow the hypothesis space for effective PIN inference. The LIN module leverages a large language model (LLM) as a reasoning engine to incorporate logical cues extracted from keystroke acoustics. As illustrated in Figure 7, the module consists of two stages: **Logic Extractor** and **Pruning Reasoner**. The Logic Extractor employs the LLM to deduce

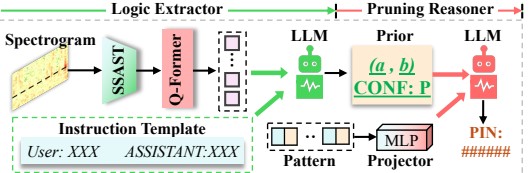

**Figure 7:** The architecture of LIN.

position-wise repetition–digit priors, while the Pruning Reasoner integrates these priors with the individual's digit-level latency patterns from the previous module to infer PINs.

**Logic Extractor.** Prior studies show that users often choose PINs with repeated digits, leading to a non-uniform distribution across positions Wang et al. (2017); Simon & Anderson (2013). Building on this observation, we design a novel **Logic Extractor** to derive position-wise repetition–digit priors from keystroke acoustics. The pipeline consists of a pretrained acoustic spectrogram encoder SSAST Gong et al. (2022), a Q-Former Li et al. (2023) as the translator, and Llama3.1-8B-Instruct Dubey et al. (2024) as the reasoning engine. Unlike a shallow classifier, the LLM serves as an extensible reasoning component, allowing auxiliary side-channel information to be incorporated via prompts to further improve PIN inference. To generate the repetition-digit priors, SSAST first extracts cues from the keystroke spectrogram. These representations are translated into continuous tokens by the Q-Former and then passed to the LLM for reasoning. To constrain the LLM to autoregressively generate strictly formatted class tokens, we adopt an instruction template as input:

> *#USER: [Continuous Query Tokens] Please choose the active repetition classes (0–15) with the associated confidence scores.*
> *#ASSISTANT: Class: a, CONF: p, . . .*

These predicted class tokens from the *Assistant* are decoded into a pre-defined upper-triangular matrix with $\binom{6}{2} = 15$ unique entries. Each entry corresponds to a specific position pair. This matrix provides a semantic representation of pairwise repetition probabilities and is used for subsequent PIN inference.

**Training Scheme of Logic Extractor.** Although this process is effective, one critical issue is that the LLM and Q-Former are primarily optimized for general-purpose semantics rather than task-specific representations Sung et al. (2022); Li et al. (2023). Consequently, without appropriate adaptation, they cannot capture the fine-grained cues required for position-wise repetition digits. To overcome this critical challenge, we adopt a two-stage training scheme consisting of a short pretraining stage followed by joint optimization.

**In the first stage,** we freeze the pretrained SSAST and LLM. Then, we train the Q-Former and its projection layers with a binary cross-entropy loss $\mathcal{L}_{\text{BCE}}^{\text{pointer}}$ for 2 epochs. The output is mapped to an upper-triangular binary matrix $P \in [0,1]^{6 \times 6}$, following the same structured representation as in the prior generation. This allows the Q-Former to distill spectrogram features into structured tokens sensitive to digit repetitions. **In the second stage,** we optimize the full pipeline end-to-end by fine-tuning the LLM with LoRA, enabling task-specific adaptation at minimal cost. Since at most three repetition-digit position pairs can occur for a standard 6-digit PIN, we formulate the task as multi-label classification. The Logic Extractor is trained with a multi-label binary cross-entropy loss on the *Assistant* segment tokens, along with the auxiliary pointer loss on the Q-Former:

$$\mathcal{L} = \mathcal{L}_{\text{BCE}}^{\text{class}} + \lambda \mathcal{L}_{\text{BCE}}^{\text{pointer}}, \tag{3}$$

where $\mathcal{L}_{\text{BCE}}^{\text{class}}$ denotes the per-class binary cross-entropy across the 15 outputs. We set $\lambda = 0.2$ to provide lightweight structural supervision to prevent the LLM from ignoring acoustic cues.

**Pruning Reasoner** The **goal** of the Pruning Reasoner is to incorporate the semantic position-wise repetition-digit priors with the recognition patterns from early modules and narrow the hypothesis space for effective PIN inference. To do this, we design the following instruction template:

> *#Prior: REP: (i,j); CONF: p  #Recognition: [$E_r$]*
> *#USER: Individual pattern and repetition prior are given. Please infer the 6-digit PIN.*
> *#ASSISTANT: PIN: [######]*

Here, **Prior** is the textual tokens of the repetition-digit priors, and $E_r$ represents the projected recognition patterns in the LLM embedding space. To ensure dimensional alignment, we use a trainable MLP to adjust the dimension of $E_r$ from the recognition patterns in advance. Instead of continuous embedding integration, this hybrid input has two main benefits: **(i)** It enables seamless integration of recognition features with position-wise repetition priors in the same semantic space, which allows the LLM to reason over them in a unified manner. **(ii)** It provides extensibility to incorporate diverse auxiliary priors or leakage cues (e.g., partial digits, user-specific biases) directly as textual instructions, thereby improving robustness and consistency under limited supervision.

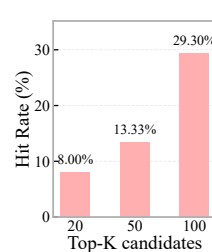

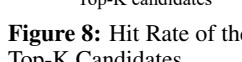

**Figure 8:** Hit Rate of the Top-K Candidates.

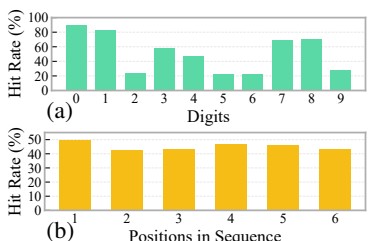

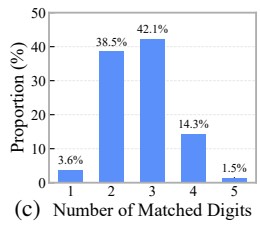

**Figure 9:** Inference performance when the ground-truth PIN is absent from the Top-100: (a) digit-wise success rate; (b) position-wise success rate; (c) maximum number of correctly matched digits within the Top-100 candidates.

**Training Scheme of Pruning Reasoner.** To align recognition features with the LLM space, we adopt a two-stage optimization. **In the first stage,** we freeze the LLM and train only the projector with cross-entropy loss on the *Assistant* tokens for 2 epochs. This prevents noisy gradients from disrupting the LLM in the early stage. **In the second stage,** we jointly train the projector and the LLM with the same cross-entropy loss using LoRA fine-tuning. This enables the LLM to integrate recognition patterns with textual repetition-digit priors and effectively narrow the hypothesis space for PIN inference.

# 7 IMPLEMENTATION AND EVALUATION

## 7.1 IMPLEMENTATION DETAILS

We evaluate *RecSpy* using a standard attacker–victim setting. The victim device is a smartphone running either Android or iOS, while the attacker uses a desktop computer for training and inference. We test three representative smartphones: Samsung Galaxy S24+ (Android 15), Samsung Galaxy S25 Ultra (Android 15), and Apple iPhone 15 (iOS 18). The attacker's software is deployed on these devices as a covert background process with microphone access. For the user study, we recruited 40 volunteers aged 18–60. Each participant entered 10 random PINs on the supplied devices while *RecSpy* silently recorded keystroke acoustics. Model training was conducted offline on an NVIDIA A100 80GB GPU. We also include a random guessing approach as our baseline.

## 7.2 TOP-K CANDIDATE HIT RATE

In Figure 8, we first evaluate the correct hit rate within the top-ranked candidate set. *RecSpy* achieves hit rates of 8.0%, 13.3%, and 29.3% for Top-20, Top-50, and Top-100, respectively. Interestingly, the per-guess hit rate is not strictly monotonic (0.40%, 0.26%, and 0.29%). This is because sample imbalance across age groups shifts some correct cases to lower ranks. Compared with the random guessing baseline of $10^{-6}$ for 6-digit PINs, these results represent probability improvements of roughly $4000\times$, $2600\times$, and $2900\times$. **In summary,** *RecSpy* elevates PIN inference from an intractable random guess to a feasible targeted attack.

## 7.3 HIT RATE OF PARTIAL INFERENCE

In this part, we evaluate the digit-wise and position-wise hit rates when the ground truth is absent in the Top-100 candidates.

● **Digit-wise Hit Rate.** Figure 9 (a) shows the digit-wise correct hit rate. The results support our previous observation that the recognition latency reflects inherent biases in visual simplicity and structural asymmetry. Specifically, digits 0 and 1 achieve the highest correct probabilities of 89.9% and 83.5%, respectively. Digits 4 and 7 exhibit higher discriminability owing to their asymmetric structure. Interestingly, since digit 8 has a near-symmetric shape, it also achieves a relatively high probability of 70.2%. The remaining digits exhibit much lower probabilities, as they are more visually confusable.

● **Position-wise Hit Rate.** Figure 9 (b) presents the position-wise correct guess probability. The success probability has relatively small variations. This indicates that recognition latency patterns are consistently exploitable across different digit positions within the PIN sequence.

● **Maximum Number of Inferred Digits.** Figure 9 (c) displays the distribution of the maximum number of digits correctly matched within the Top-100 candidate PIN sequences. The results show that even when the exact ground-truth PIN is absent, a large portion of the candidates still contain

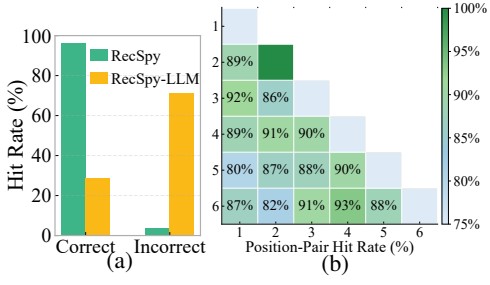

**Figure 10:** Hit rates of *RecSpy* in inferring repeated digits: (a) overall performance on position pairs with/without LLM reasoning, (b) position-wise results.

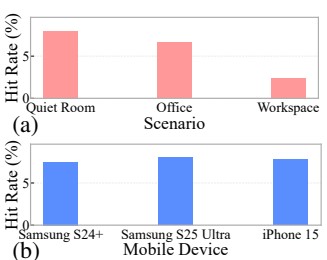

**Figure 11:** Top-20 Candidate Hit Rates Under (a) different Scenarios and (b) different Mobile Devices.

partial matches. The majority of sequences can match three or more digits, and in some cases up to five digits are correctly inferred. **In summary,** *RecSpy* can significantly reduce the uncertainty of PIN inference and achieve effective attacks on legitimate users.

### 7.4 Repeated Digit Hit Rate on Position Pairs

• **Ablation Evaluation.** To evaluate the contribution of the LLM-based reasoning component, we design a ***RecSpy*-LLM** approach to directly infer repeated digits from an individual's digit-level recognition latency pattern. This approach does not use the LLM for prior extraction and logical reasoning. Figure 10 (a) reveals a clear performance decline, with the hit rate for inferring repeated digits across position pairs dropping from $95\%$ to $30\%$.

• **Position-wise Repeated-digit Hit Rate.** In Figure 10 (b), we also evaluate the position-wise hit rate for repeated digits, where the horizontal and vertical axes represent the digit positions. Each cell denotes the probability that a repeated digit at the given position pair is correctly inferred. The results show consistently high hit rates across all position pairs, with most values exceeding $80\%$, which indicates that *RecSpy* can reliably infer repeated digits regardless of their relative positions. **In summary,** the reasoning component of *RecSpy* is crucial for leveraging the hidden repetition prior to narrow the effective search space for PIN inference.

### 7.5 Generalization to Devices and Environments.

• **Cross-Environment Performance.** We evaluate the robustness of *RecSpy* under various indoor environments with increasing ambient noise, including a quiet room, a shared office, and an open workspace. Figure 11 (a) displays the Top-20 candidate hit rate in these scenarios. The results show that *RecSpy* remains effective in quiet and moderately noisy conditions, with hit rates of $8.00\%$ in the quiet room, $6.63\%$ in the office, and $2.40\%$ in the noisy workspace. **Although these numbers may appear modest, they represent a practical threat in real settings: once a malicious app is installed, an attacker can continuously collect data and attempt guesses over time.** Therefore, we believe that *RecSpy* is relatively robust under noisy scenarios. This is because *RecSpy* mainly relies on keystroke acoustics to identify *temporal* interval patterns, rather than performing spatially precise acoustic localization, making it less dependent on denoising or high-fidelity signal recovery.

• **Cross-Device Performance.** We further evaluate the cross-device performance of *RecSpy*. As shown in Figure 11 (b), hit rates are largely consistent. **In summary,** *RecSpy* can achieve effective attacks across diverse scenarios and devices.

## 8 Conclusion

In this paper, we presented *RecSpy*, a novel cognition-driven acoustic side-channel attack that infers PINs on randomized soft keyboards. By modeling digit recognition latency through contrastive and self-supervised learning, and by introducing a Logic-Guided Inference Network that leverages LLM-based reasoning, *RecSpy* effectively exploits human cognitive patterns for PIN inference. Our evaluation on both Android and iOS devices demonstrates that *RecSpy* can improve inference success rates by up to 4000× compared with random guessing, which demonstrates severe security issues to mobile authentication. Moreover, our results reveal that advances in representation learning and LLMs can be misused to compromise security and privacy.

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
