# OpenReview forum: "RecSpy: Cognition-Driven PIN Inference on Randomized Soft Keyboards"
_ICLR.cc/2026/Conference — ICLR 2026 Conference Withdrawn Submission_

### Official Review · Reviewer_nroz · 2025-10-29

**Soundness:** 2
**Presentation:** 3
**Contribution:** 3
**Rating:** 4
**Confidence:** 3

**Summary:**

Authors present RecSpy, a novel cognition-driven acoustic side-channel attack that infers PINs on randomized soft keyboards. RecSpy exploits distinct human recognition latencies for symbolic numbers. They model cognitive latency patterns and leverage acoustic keystroke signatures using contrastive and self-supervised learning to learn individual and digit-level recognition features. The authors also introduce a novel Logic-Guided Inference Network that integrates recognition patterns with the reasoning capabilities of a LLM to prune the hypothesis space and infer complete PIN sequences. The method is tested against Android and iOS devices. The method improves the probability of a successful PIN inference by up to 4000x compared to random guessing.

**Strengths:**

- **Novel focus on randomized soft keyboard for PIN entry**, which is commonly used as a countermeasure against fixed-layout attacks in practice. This makes the PIN inference problem harder than that on fixed-layout but more realistic and vital to investigate.
- **Novel concept of exploiting human cognitive recognition latency** for inferring PIN digits. This is novel and very interesting concept that also seems to be supported by referenced work. This approach enables a realistic threat model that only assumes microphone access through an installed malicious app on the victim's phone.
- The attack seems to generalize between different smartphone and OS platforms (Android and iOS).

**Weaknesses:**

- **The attack does not seem to be very successful.** Hit rates of 8%, 13%, and 29% for Top-20, Top-50 and Top-100, respectively, do not seem all that successful. Moreover, PIN entry usually has limits of 3 to at most 5 tries before being blocked by the apps (bank account apps or similar), making this very unrealistic.
- **Lack of justification and evidence-supported rationale for the steps and components of the proposed method.** While the paper includes includes an ablation study for the third part using LLMs, the rest of the process offers only vague or no reasoning for design choices. Each step of the method is packed with many different components (e.g., SENs, CNNs, LSTMs in just one part of the method). The selection of these specific components appears arbitrary, as the authors provide little to no justification or empirical support for their choices. This raises significant doubt regarding the necessity for such highly complex method, and makes the method challenging to evaluate and validate.
- **The provided explanation of the method is incomplete,** leaving some open questions about its overall process. Specifically, there is ambiguity surrounding the collection and application of user (victim) data. The authors fail to clarify when this data is collected in the process and how it is integrated and utilized by the method. The training schemes (paragraphs) require a more detailed discussion on what data is used and the precise data collection methodology. Moreover, a significant ambiguity exists in the method's ability to differentiate keystrokes: it is not clearly explained how the system distinguishes between inputs related to PINs and general smartphone usage, such as typing messages.
- **The evaluation of the proposed attack method is significantly limited** due to its comparison against only random guessing. This minimal baseline restricts the proper assessment of the attack's true performance and efficacy.

**Questions:**

These questions are, in most part, addition to the stated weakness.

- My impression is that the results are not very good. Thus, what are the reasons for such a performance, and what makes these metrics and these results indicate a successful attack and a real threat to user's PIN security? Top-20 does not seem to be good enough for limited PIN tries, and with the statement from the paper, "an attacker can continuously collect data and attempt guesses over time", the question raises as to how long does it then take, and how much data is necessary?
- What are the implications of this work? Should smartphone designers/app designers do something and should users of those smartphone be concerned? This is also connected with the previous point on the attack's success.
- The terms utilized in this work, such as Individual-level Recognition, and Digit-level Discrimination, individual-specific patterns and digit-level latencies are not very clear. Moreover, they are never explicitly explained. For example, only in training scheme for IRM, it is said that the data is labelled with the age groups, meaning that this is where the reader learns that the individual-level recognition means the age group recognition specifically (if I understand that correctly). So, my suggestion is that these terms are explained more explicitly to avoid confusion.
- Why are all three steps of the method necessary? How do you select all the components and their combinations for each of the steps? Also, some of the hyperparameters are mentioned, such as number of epoch, and the choice for those is also not discussed.
- Another confusion stems from the data and data handling.
	- There is a dataset in Section 3.3. (143 participants) and a dataset from Section 7.1 (40 participants). Is the 40 participants data a a subset of 143 participants data or a completely new group of participants? With the same group of people, there can be bias in the ML methods utilized.
	- Is it an assumption that the attacker has a similar dataset that is collected offline and labelled with age groups?
	- At which point is the victim's data collected (for inference) and how is it utilized in the method?
	- The data described in Section 3.3 is preprocessed by removing silence and aligned to a specific time. Is the collected data from victim user also preprocessed in this way?
	- From the collected victim's data, how does the attacker differ between keystrokes related to the PIN entries and the rest of the smartphone usage?
- Why was the attack method only compared with random guessing?

Additional comments/questions (less relevant)
- How does your approach work if the keyboard layout is not randomized?
- "but the relative differences among digits remained unchanged." - This is not true, look at times for numbers 2 and 8 in the 31-40 and 41-50 age group.
- What would be limitations of this work, or possible countermeasures?
	- Can users detect suspicious background operations, e.g., from the decreasing battery life due to more power used while they are not actively using their smartphone?
- Due to the high complexity of the method, reproducing the work appears tedious and is unlikely to be successful given the lack of reported hyperparameters. Thus, providing the source code and, ideally, the collected datasets would significantly benefit this research direction and allow for proper validation.

---

> ### Author Response · Authors · 2025-11-20
> **Response to Reviewer nroz**
>
> We appreciate your comments and advice. Our corresponding responses and revision plans are as follows:
>
>
> R1: Sorry for the confusion. First of all, our work focuses on widely occurring real-world scenarios rather than being restricted to isolated single-user cases. Second, regarding limited PIN attempts, once microphone permission is granted, the malicious app can continuously and passively collect data when the user interacts with the device. The attacker can entry PINs over multiple days without triggering lockouts. In practice, about ten days of routine PIN usage are sufficient for our model to infer the PINs. Therefore, our attack is feasible and reveals a practical threat under typical retry-limit policies.
> Again, we want to emphasize that our threat model highlights widespread real-world risks involving numeric PIN leakage on networked systems, which follow the scenarios reported by the FBI IC3 (2024 Internet Crime Report) and the CISA (Implementing Phishing-Resistant MFA).
>
>
> R2: Our work reveals broad practical security risks and highlights the need for stronger platform-level defenses and indicates that user-interaction traces should be properly encrypted to prevent sensitive data leakage. It also exhibits the LLM's potential as the reasoning component. Smartphone designers may adopt more persistent sensor-usage indicators and tighten third-party installation policies against passive background attacks. Users should be cautious when granting permissions to untrusted apps and avoid relying on highly predictable interaction habits.
>
>
> R3: Yes, Individual-level Recognition extracts age-related recognition patterns. We apologize for any confusion and will clarify these terms in revision.
>
>
> R4: The three modules are designed for reliable inference: IRM extracts age-related recognition features, DDN details them with digit-level discriminative patterns, and LIN refines the final inference from signal cues. Figure 10(a) shows that removing LIN significantly reduces accuracy, while IRM and DDN are necessary for feature extraction. Hyperparameters are selected through evaluation and validation. We will clarify these details in revision.
>
>
> R5: The datasets in Sections 3.3 and 7.1 are collected from entirely distinct participant group, and each dataset is balanced across age groups. We will clarify this in the revision.
>
>
> R6: We do not assume that the attacker must use the same training dataset as ours. As long as the attacker can collect an offline dataset from legitimate users with age-group labels, the model can be trained to capture the corresponding latency patterns. We will clarify this in the revision.
>
>
> R7: The victim’s data is collected passively during PIN entry, with the background app using the microphone. Each entry generates a short sequence of stereo tap acoustics that is uploaded to the attacker’s device, where the model ranks candidate PINs.
>
>
> R8: Yes, the victim's data will also be preprocessed before being fed into the model.
>
>
> R9: PIN-entry taps are distinguishable since they occur as a short, consecutive burst of similar tap acoustics, while interactions, such as scrolling or swiping produce irregular acoustic patterns. Based on this, the attacker can locate the coarse PIN-entry region and then apply a backward offset from the first detected tap, using the maximum expected reaction-latency bound to truncate the signal window for inference.
>
>
> R10: To the best of our knowledge, this is the first work that focuses on PIN eavesdropping from randomized keypad layouts. Prior work relies on the fixed mapping between digits and physical positions for inference, while randomized layout removes this mapping entirely. Therefore, these approaches cannot extract meaningful cues and are not applicable as baselines for our setting.
>
>
> R11: If the layout is fixed, our model can easily distinguish different PIN digits, as acoustic signal reflects both spatial mappings and cognitive latency patterns of digits. We also want to emphasize that one of the main contributions of our work is that we consider a practical scenario where the keypad layout is randomized.
>
>
> R12: Thank you for pointing this out. We will revise it accordingly in revision.
>
>
> R13: Unusual battery drains may not be reliable since it varies widely across apps and system conditions. A more practical approach is for software developers to introduce visual variationsin keypad elements (e.g., varying font sizes or colors) to disrupt the digit-level cognitive mapping exploited by our attack.
>
>
> R14: We apologize for the oversight and will release the code at a later stage. We also want to clarify that the implementation of this work is fast because: (1) the training dataset is relatively small, (2) the LLM requires no architectural modification, and (3) microphone access is super easy to obtain, as a report by CyberNews shows that thousands of legitimate apps routinely request and receive microphone permissions even when unnecessary.

---

> > ### Comment · Reviewer_nroz · 2025-11-25
> >
> > Thank you for your answers and clarifications. I have no further questions, just a few more comments based on the provided answers.
> >
> > R1: Given the clarifications, the experimental results would benefit from including the time and/or the amount of data required to achieve the reported results. This information is currently missing (if I'm not mistaken), and it would help put the attack's efficiency into perspective.
> >
> > R9: This process is not described in the paper, but it appears relevant to the success of the attack. So, it should be added and perhaps also evaluated in the paper (or a reference should be provided with supporting evaluation and description if it is a standard way to filter out PIN inputs).
> >
> > R10: I understand none of the existing methods are directly comparable with your attack, but relying only on random guessing as a baseline is insufficient. Assuming that methods for fixed layouts have better results since they have more information to work with, it would be nice to know where your results fall between random guessing (worst results) and existing methods on fixed layout (best results). This can be just a discussion on the results reported by the existing work on fixed-layout keyboards and your results.

---

> ### Author Response · Authors · 2025-11-25
> **Follow-up Clarifications for R1, R9, R10**
>
> We appreciate your constructive advice and address each point as follows:
>
> **R1:** Yes, the amount of data and time required to achieve the experimental results would meaningfully increase the efficiency of the attack. These measurements were collected during our evaluations and are already available on our side, but they were not included in the current manuscript due to space constraints. We will include them in the revised manuscript to more clearly demonstrate the practicality of our attack.
>
> **R9:** Thank you for pointing this out. We will add a clear description of this process in revision.
>
> **R10:** We do have the results on fixed-layouts ready, but due to space limitations they are not presented in the current version. We will incorporate the key figures and include a clear discussion in the revision to show the performance level of our attack relative to existing fixed-layout approaches.

---

### Official Review · Reviewer_pDWx · 2025-10-31

**Soundness:** 3
**Presentation:** 2
**Contribution:** 3
**Rating:** 4
**Confidence:** 3

**Summary:**

The paper proposes an innovative cognitive-driven acoustic side-channel attack scheme called RecSpy, which records stereo tap acoustics on smartphones and exploits digit‑specific cognitive latencies via IRM, DDN, and an LLM‑based LIN to infer 6‑digit PINs on randomized keypads. Cross‑platform tests report strong Top‑K hit rates and "up to 4000x" gains over random guessing, indicating a practical threat.

**Strengths:**

1. Novel, cognition‑driven angle leveraging digit‑wise recognition latencies and age trends
2. Well‑structured pipeline: IRM (contrastive + frequency weighting), DDN (self‑supervised, conditioned alignment), LIN (LLM‑guided repetition priors and pruning).
3. Evaluations across multiple platforms (Android/iOS), devices, and environments (quiet/noisy) demonstrate the generalizability of RecSpy.

**Weaknesses:**

1. Lack of discussion on the influence of real-world variables.
2. Per‑subject PIN samples are limited (10 each), which may constrain robustness under fully randomized layouts.
3. No comparison to lightweight baselines or reporting of latency/energy for the LLM stack.

**Questions:**

1. The assumption of cognitive delay in the paper lacks sufficient verification. How will real-world variables, such as user input habits (e.g., typing rhythm, hesitation behavior), user's physical state (fatigue, tension), device hardware response delay, etc., affect cognitive delay, and thereby blurs the core basis for the model's inference?
2. In section 3.3, the paper mentions "143 participants" for latency observation, and in section 7.1, it mentions "40 volunteers" for evaluation. Are the training and evaluation participants entirely distinct?
3. While the paper compares RecSpy to random guessing, it does not benchmark against lightweight models (e.g., logistic regression, CRFs) or prior keystroke eavesdropping techniques. Is it possible that prior methods or lighter models can achieve attack performance slightly worse than RecSpy, but with much lower cost and constraints?
4. In section 6.3, the paper uses the premise of "frequently used duplicate digits" and then uses SSAST+Q‑Former+LLM to generate the "position pair duplicate prior matrix". What are the interpretable cues in the acoustic tokens that LLM uses to derive the "duplicate positions"? Is there a risk of overfitting to the training set's PIN distribution?
5. Although Section 7.2 provides Top‑K metrics and Section 7.5 further discusses "sustaining collection and continuous attempts", it does not  take into account typical retry limits or lockout strategies in real-world systems. What is the actual success rate in real-world systems (e.g., typically 3-10 attempts at most).
6. Figures 8/9/10/11 only show average results. What is the variance across participants?
7. There are some writing and formatting errors in the paper, for example:
    * The caption of Fig 3 only mentions "Aged 51 to 60", whereas Fig 3 displays four age groups.
    * A closing parenthesis ")" is missing in the right-hand label of Fig 3.
    * On line 363, "Pruning Reasoner" -> "Pruning Reasoner."

---

> ### Author Response · Authors · 2025-11-20
> **Response to Reviewer pDWx**
>
> We appreciate the valuable comments and advice. The corresponding explanations and revision plans are as follow:
>
> **R1**: Sorry for any confusion. During our evaluation, participants of different ages entered their PINs under various scenarios. In other words, we did not strictly force them to follow the same behavior patterns when entering their PINs. We will further clarify the scope and assumptions of the threat model in the revised version.
>
> **R2**: Yes, the training and evaluation participants are completely independent. We apologize for the unclear presentation and will make this explicit in the revision.
>
> **R3**: Randomized layouts eliminate the position–digit correspondence that prior works rely on. Without spatial features, these approaches CANNOT extract meaningful cues and therefore are not applicable as baselines in this setting.
>
> **R4**: Duplicate digits produce highly similar acoustic embeddings because they arise from the same tap geometry within a single PIN entry. The SSAST–Q-Former–LLM pipeline leverages these within-entry acoustic similarities as cues to form the position-pair duplicate prior.
> This process does not expose the model to full PIN values or digit-frequency statistics. The labels only indicate whether two taps belong to the same digit instance. As a result, the pipeline cannot memorize global PIN-distribution patterns and thus does not overfit PIN frequencies.
>
>
> **R5**: Our threat model assumes that an attacker can leverage daily retry-limit resets by making a small number of attempts each day. Many real-world systems clear failed-attempt counters after a successful login by the legitimate user, which allows sustained low-frequency guessing without triggering lockouts. We will incorporate the corresponding real-world success rates under typical retry limits (0.75% for top 5, and 2.32% for top 10) into the revised evaluation.
>
> **R6**: The variance varies across participants and is age-dependent. Younger users have lower Top K hit rates, while the 41–60 groups show higher values due to their more pronounced cognitive latency gaps. We will include the variance statistics for each age group and the corresponding analysis in the revised version.
>
> | Age Group | Top-5 Hit Rate | Top-10 Hit Rate | Top-20 Hit Rate |
> |----------------|---------------------|----------------------|-----------------------|
> | 18–30        | 0.36%              | 1.13%               | 3.89%                |
> | 31–40        | 0.64%              | 1.97%               | 6.78%                |
> | 41–50        | 1.03%              | 3.19%               | 10.98%              |
> | 51–60        | 1.09%              | 3.36%               | 11.60%              |
>
>
> **R7**: We apologize for the oversights and will correct all writing and formatting issues in the revised version.

---

### Official Review · Reviewer_QF7j · 2025-11-01

**Soundness:** 2
**Presentation:** 2
**Contribution:** 2
**Rating:** 2
**Confidence:** 3

**Summary:**

The paper proposes *RecSpy*, an acoustic side-channel attack for inferring PINs entered on randomized mobile soft keyboards by combining cognitively motivated digit-recognition latency patterns with spectrogram features and a two-stage reasoning pipeline. It introduces the IRM to learn user-specific recognition embeddings, the DNN to derive digit-level latency cues without explicit per-digit labels, and the LIN that uses an LLM to extract repetition priors and prune candidate PINs. The method targets a practical threat model in which a microphone-permitted background app records stereo keystroke acoustics on Android and iOS and performs offline inference, with evaluations across devices and indoor environments.

**Strengths:**

- The paper targets randomized soft keyboards and articulates a microphone-only threat model that avoids privileged sensors or external hardware.
- The architectural choices for the proposed modules are clearly described.
- The evaluation spans multiple devices and indoor settings and analyzes partial inference and repeated-digit priors.

**Weaknesses:**

- In this paper, the security significance is framed largely as a multiplicative gain over random guessing rather than attacker-centric success metrics (e.g., partial/average-case guessing entropy under retry limits). while Bonneau et al. recommend evaluating authentication schemes explicitly under throttled vs. unthrottled guessing to reflect real attack costs [1]. Reporting only something like "x improvement over $10^{-6}$" (e.g., the numbers reported in Section 7.2) can be somewhat misleading.

- Given the randomized layout, the paper should include a direct comparison against an audio-only baseline, such as a layout-agnostic KeyListener-like model [2] retrained to either classify each tap's spectrogram into digit labels without position cues or infer same/different constraints across taps, so the unique contribution of the cognition-driven modules is clear.

- Reliance on LLM reasoning to extract/prune repetition priors raises robustness and faithfulness concerns. Prior work shows chain-of-thought style reasoning can be brittle and unfaithful, and performance often depends on decoding strategies like self-consistency [3], which suggests the Pruning Reasoner may be sensitive to prompts/seeds and could overfit to training distributions unless carefully stress-tested.

- There are also reproducibility issues: the paper provides no stated plan to release code or data, and that the dataset is only described at a high level (devices, participant counts, and collection protocol) without an accessible corpus or retrieval details, preventing independent verification or re-analysis.

## References

- [1] J. Bonneau et al. The Quest to Replace Passwords. IEEE S&P 2012.
- [2] L. Lu et al. KeyListener: Inferring Keystrokes on QWERTY Keyboard of Touch Screen through Acoustic Signals. IEEE INFOCOM 2019.
- [3] X. Wang et al. Self-Consistency Improves Chain-of-Thought Reasoning in Language Models. ICLR 2023.

**Questions:**

- Under your stated threat model, what concrete evidence shows background audio capture is feasible during real PIN entry on modern Android/iOS?
- How sensitive is performance to device, microphone placement, environmental noise, and keypad randomization frequency? Where does the system fail?
- If code or data cannot be released, what exact artifacts (trained weights, prompts, logs, synthetic generators, evaluation scripts) will you provide to enable independent replication?

---

> ### Author Response · Authors · 2025-11-20
> **Response to Reviewer QF7j**
>
> Thank you for your comments.
>
> **We would like to provide the following clarifications to the concerns noted in the weakness section:**
>
> **R1**: Sorry for the confusion. Our goal is to quantify the attacker’s advantage in a population-scale PIN-entry scenario. Therefore, the attacker can safely try a few candidates each day, since daily retry resets allow small numbers of safe guesses without triggering lockouts. Figure 8 evaluates success probability in a throttled setting by using the top-k hit rate as a metric, which reflects attack costs under both constrained and repeated daily guessing conditions. We will explain this in the evaluation section.
>
> **R2**: It has been proven that randomization will break the mapping between digit and physical key position, which makes the KeyListener-like baselines inapplicable. Therefore, our novel cognition-driven module is specifically designed to leverage the cognitive relationship between digits and tap-timing patterns to achieve PIN inference on layout randomization.
>
> **R3**: Our reasoning module does not rely on open-ended chain-of-thought. This is because the pipeline operates on acoustic embeddings and a fixed deterministic prompt rather than free-form textual reasoning, which makes open-ended CoT neither necessary nor applicable. Since it ranks or prunes candidate PINs based on observed patterns, without generating intermediate logical steps, its behavior is far less sensitive to prompts or sampling variability than general open-ended reasoning.
>
> **R4**: We apologize for overlooking the artifact option. We will provide the code and necessary replication materials in the revision stage.
>
>
> **The corresponding responses to the questions are as follow:**
>
> **R1**: As discussed in the Introduction section, modern Android and iOS allow foreground apps with microphone permission to capture audio while running in the background, such as VoIP apps, voice-note utilities, and certain accessibility services. Prior work has demonstrated that tap and acoustic events can be reliably recorded under these conditions on both platforms. Therefore, our threat model reflects a practical scenario, as PIN entry commonly occurs while such apps remain active.
>
> **R2**: As demonstrated in Figure 1, microphone placements on modern mobile devices are standardized. In addition, we have evaluated the cross-device robustness and the impact of environmental noise in Figures 11 (a) and (b), respectively. Moreover, in our design, keypad randomization frequency does not affect our inference, since our attack does not rely on spatial digit–key mappings. Instead, it exploits intrinsic cognitive timing patterns of digit recognition, which remain consistent across layout changes.
>
> **R3**: As we mentioned above, we will provide the code and corresponding materials in the revision stage.

---

> > ### Comment · Reviewer_QF7j · 2025-11-24
> >
> > Thanks for the response. Regarding the third point, even without a chain-of-thought process, LLM outputs can still be random. I'd like to know how to design a prompt that makes an LLM-based reasoning engine behave more robustly?

---

> > > ### Author Response · Authors · 2025-11-24
> > > **Follow-up Clarification for R3**
> > >
> > > Thank you for the question. We address the concern with the following strategies. **First,** we designed fixed response templates to tighten the prompt structure. For the Logic Extractor stage, the prompt defines all possible choices (15 classes) to strictly constrain the LLM to output sequences of formatted, symbolic decision tokens (e.g., Class: a, CONF: p, ...). For the Pruning Reasoner stage, the prompt requests the LLM to output six-digits PINs. These templates ensure that the target is highly constrained and minimal, so the LLM’s generation entropy is inherently low to be less sensitive to randomness. **Second,** we utilize a conservative decoding approach to suppress stochastic sampling and prioritize maximum likelihood decisions. This approach makes the output stable and consistent across multiple runs. **Third,** in this context, the prompt's function is not to guide complex logic, but to achieve a highly deterministic mapping between the input features and the formatted output tokens. Therefore, it is sufficiently robust for our inference refinement. In practice, the LLM can **consistently** generate the required information using these prompts.

---

> > > > ### Author Response · Authors · 2025-11-25
> > > > **Supplementary Explanation on Reducing LLM Randomness**
> > > >
> > > > In addition, we would like to clarify that we have carefully designed our training scheme to mitigate the inherent stochasticity of LLM-based reasoning. Specifically, our elaborated two-stage training pipeline is explicitly constructed to minimize randomness and stabilize the model’s inference behavior. This is challenging because it requires carefully coordinating multiple components and ensuring that each stage produces representations precise enough for the next module to reliably rely on. **In the first stage,** we freeze the LLM and optimize only the Q-Former and projector components to ensure that the LLM receives stable, low-variance embedding representations rather than noisy acoustic features. This removes upstream randomness and forces the model into a consistent and well-structured input space. **In the second stage,** we fine-tune the LLM using structured multi-label BCE supervision on fixed-format class tokens to guide the model away from open-ended generation. This supervision constrains its autoregressive behavior to a narrow, well-defined output space, which ensures consistent and predictable decoding. Finally, we use LoRA-based adaptation to update only a small subset of parameters, preventing drift while still enabling task-specific reasoning. Through these carefully orchestrated steps, we significantly decrease the stochastic behavior of a general-purpose LLM and enable it to function as a controlled, low-variance reasoning module.
> > > >
> > > > To further support reproducibility, in the final version we will release the complete source code together with a step-by-step guide that provides environment-specific instructions to accommodate different user systems and configurations.

---

> > > > > ### Comment · Reviewer_QF7j · 2025-11-28
> > > > >
> > > > > Thanks for the detailed response. After reading it, my impression is that constructing the Logic Extractor is a fairly complex process and that it plays a central role in the proposed RecSpy framework. However, in the current version of the paper, this crucial component is only described briefly.
> > > > >
> > > > > For a future version, I would strongly encourage the authors to provide a more comprehensive description of the Logic Extractor, including details about the training dataset, the specific LLM used, and the key hyperparameters employed for fine-tuning.
> > > > >
> > > > > Based on the additional clarification provided, I have decided to increase my score. That said, I still feel that the response is primarily qualitative and lacks supporting quantitative results. I hope the authors will further investigate and elaborate on the design and evaluation of the Logic Extractor in future work.

---

> > > > > > ### Author Response · Authors · 2025-11-28
> > > > > > **Response to Reviewer QF7j**
> > > > > >
> > > > > > Thank you for your constructive feedback. In the revised version, we will provide the full details of the Logic Extractor and release all related datasets and code to ensure transparency.

---

### Note · Authors · 2026-01-25

**Comment:**

Due to the submission plam and scheduling constraints, as well as the review comments received, we have decided to withdraw the paper.

**Withdrawal Confirmation:**

I have read and agree with the venue's withdrawal policy on behalf of myself and my co-authors.